# Direct observation of the complex S(IV) equilibria at the liquid-vapor interface

Tillmann Buttersack [1,8] ✉, Ivan Gladich [2,8] ✉, Shirin Gholami[1], Clemens Richter[1], Rémi Dupuy [3], Christophe Nicolas[4], Florian Trinter [1], Annette Trunschke[1], Daniel Delgado [1], Pablo Corral Arroyo[5], Evelyne A. Parmentier[5], Bernd Winter [1], Lucia Iezzi[6], Antoine Roose [6,7], Anthony Boucly[6], Luca Artiglia [6], Markus Ammann [6], Ruth Signorell[5] & Hendrik Bluhm [1] ✉

The multi-phase oxidation of S(IV) plays a crucial role in the atmosphere, leading to the formation of haze and severe pollution episodes. We here contribute to its understanding on a molecular level by reporting experimentally determined $pK_a$ values of the various S(IV) tautomers and reaction barriers for $SO_2$ formation pathways. Complementary state-of-the-art molecular-dynamics simulations reveal a depletion of bisulfite at low pH at the liquid-vapor interface, resulting in a different tautomer ratio at the interface compared to the bulk. On a molecular-scale level, we explain this with the formation of a stable contact ion pair between sulfonate and hydronium ions, and with the higher energetic barrier for the dehydration of sulfonic acid at the liquid-vapor interface. Our findings highlight the contrasting physicochemical behavior of interfacial versus bulk environments, where the pH dependence of the tautomer ratio reported here has a significant impact on both $SO_2$ uptake kinetics and reactions involving $NO_x$ and $H_2O_2$ at aqueous aerosol interfaces.

An in-depth understanding of the physicochemical processes in the atmosphere is essential for improving the reliability and predictive capability of air quality and climate models, where the sulfur cycle plays an essential role. Sulfur enters the Earth's atmosphere from natural and anthropogenic sources[1]. An important contributor is sulfur dioxide ($SO_2$) as the main source of S(IV) species, with local $SO_2$ concentrations strongly dependent on the geographical location[1].

Sulfuric acid ($H_2SO_4$) is the major acidifying species in the atmosphere and thus provides an important contribution to aerosol, cloud-droplet, and rain-water acidity[2]. The oxidation of S(IV) species to sulfuric acid can proceed through the reaction of $SO_2$ with OH radicals in the gas phase. In aqueous environments, such as aerosol particles and cloud droplets, the hydrolysis products of $SO_2$ can react with peroxides, ozone ($O_3$), and nitrogen dioxide ($NO_2$), resulting in the formation of several distinct aqueous S(IV) species. The concomitant reduction of some of the oxidants (e.g., reduction of $NO_x$ to $HNO_2$) during these reactions is important for the cycling of halogens[3,4] or nitrogen oxides[5] in the atmosphere. Among the multi-phase oxidation pathways, the reactions between S(IV) and both $NO_2$ and $H_2O_2$ are of particular importance since the kinetics of these reactions appear to be enhanced under the high ionic strength conditions of aqueous aerosol particles[6–8]. This enhanced kinetics (which is also influenced by the pH of the aerosol) has been suggested to cause the unexpectedly large increase of aerosol mass in haze events in polluted urban

[1]Fritz Haber Institute of the Max Planck Society, Faradayweg 4–6, 14195 Berlin, Germany. [2]Qatar Environment and Energy Research Institute, Hamad Bin Khalifa University, P.O. Box 34110, Doha, Qatar. [3]Sorbonne Université, CNRS, Laboratoire de Chimie Physique-Matière et Rayonnement, Paris Cedex 05 F-75005, France. [4]Synchrotron SOLEIL, L'Orme des Merisiers, Saint-Aubin-BP 48, 91192 Gif-sur-Yvette, France. [5]Laboratory of Physical Chemistry, Department of Chemistry and Applied Biosciences, ETH Zürich, Vladimir-Prelog-Weg 2, Zürich CH-8093, Switzerland. [6]PSI Center for Energy and Environmental Sciences, Paul Scherrer Institute, Villigen PSI CH-5232, Switzerland. [7]Present address: IMT Nord Europe, Institut Mines-Télécom, University Lille, Lille F-59000, France. [8]These authors contributed equally: Tillmann Buttersack, Ivan Gladich. ✉e-mail: buttersack@fhi.mpg.de; igladich@hbku.edu.qa; bluhm@fhi.mpg.de

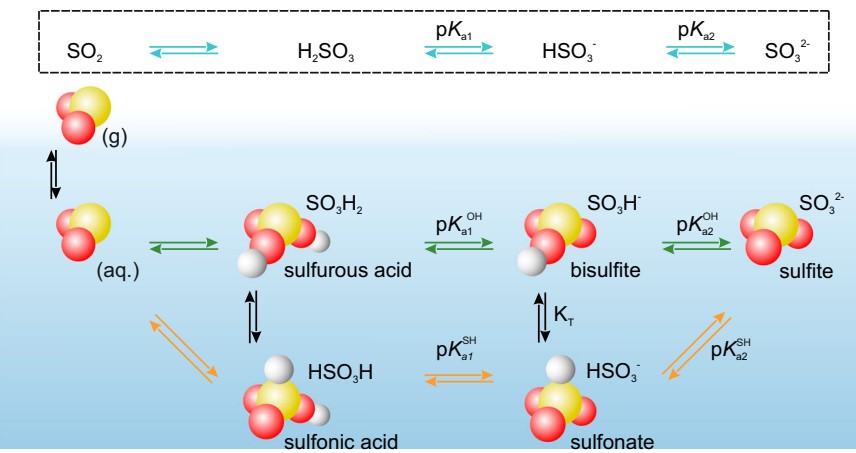

**Fig. 1 | Acid–base equilibria of S(IV) species at the liquid-vapor interface.** The acid–base equilibria at the interface, including the tautomeric forms, are investigated by photoelectron spectroscopy and are compared to the bulk equilibria, which are determined by Raman spectroscopy.

environments[6,8]. A significant number of studies have focused on the hydrolysis kinetics of $SO_2$[9–12] and provided substantial evidence that the formation of a surface complex is the limiting step for uptake and hydrolysis during the adsorption of $SO_2$ from the gas phase on aqueous solution interfaces. This suggests that there is a significant surface coverage by $SO_2$ under equilibrium conditions[10,13–16]. There is also evidence that surface processes are involved in the enhancement of the reactivity of the S(IV) oxidation processes in aerosol particles mentioned above.[8] Therefore, the accurate knowledge of the surface propensities and surface equilibria of S(IV) species is of substantial importance.

Hydrolysis of $SO_2$ initially leads to the formation of $H_2SO_3$ (see Fig. 1) in the form of sulfurous ($SO_3H_2$) or sulfonic ($HSO_3H$) acids, which are difficult to detect because of their short lifetimes (in the tens of $\mu$s range) in the presence of water[3,17]. The acids can be deprotonated twice[3], in the first step leading to the formation of the tautomers bisulfite ($SO_3H^-$) and sulfonate ($HSO_3^-$), and in the second step to form sulfite ($SO_3^{2-}$), see Fig. 1. The tautomer ratio $K_T$

$$SO_3H^- \rightleftharpoons^{K_T} HSO_3^- \tag{1}$$

in the bulk solution was investigated theoretically[18] and experimentally[19–21] (e.g., Raman, nuclear magnetic resonance, and X-ray absorption spectroscopy) for low pH and as a function of temperature. It was found that at room temperature, this equilibrium favors bisulfite, while it shifts to sulfonates at higher temperatures[20,21].

An open question is still the tautomer ratio and the lifetime of these species at the liquid–vapor interface, where heterogeneous reactions take place. Acid–base equilibria[22,23] and the concentration of ions[24,25] at the liquid–vapor interface can differ from those in the bulk solution, with consequences for heterogeneous reaction rates and mechanisms.

In the present work, we use X-ray photoelectron spectroscopy (XPS) and Raman spectroscopy, as well as MD-simulations and BE calculations to monitor the complex equilibria of S(IV) species (see Fig. 1) at the liquid–vapor interface and in the bulk of the solution. We have determined the tautomer ratio as a function of pH and the $pK_a$ values of the tautomeric forms, as well as, for the first time (to the best of our knowledge), experimentally observed dissolved $SO_{2(aq.)}$ at the solution-vapor interface. Our experimental results are supported by MD simulations, which, in addition, provide details on the competitive formation of sulfurous and sulfonic acid through the hydrolysis of $SO_2$ or the protonation of bisulfite and sulfonate.

## Results and discussion

Selected S $2p$ XPS spectra of 400 mM sodium sulfite solutions with a pH ranging from 10.4 to 0.36 are depicted in Fig. 2. The estimated probing depth into the solution in the XPS experiments is of the order of 2 nm. All photoelectron binding energies (BEs, referenced to the HOMO of liquid water[26]) and other fit parameters are displayed in Table 1. At pH = 10.4 only one peak due to the sulfite ion is observed. At the bulk $pK_a$ of pH = 7 (Fig. 2b), two additional peaks at higher BEs are evident in the S $2p$ spectrum, which account for about 50% of the total signal. These peaks are attributed to the tautomers bisulfite and sulfonate. A spectrum of a methane sulfonate solution confirms the assignment of the sulfonate peak (see Supplementary Fig. 2).

At pH = 4, the bisulfite and sulfonate peaks dominate the spectrum (Fig. 2c). In addition, a narrow $2p$ doublet appears at higher BE, which we attribute to gas-phase $SO_{2(g)}$ (purple). With decreasing pH (1.8, Fig. 2d), the $SO_{2(g)}$ signal increases, while the total S $2p$ intensity decreases. In addition, the relative intensity of sulfonate vs. bisulfite changes in favor of sulfonate, until in the spectrum at pH = 0.8 (Fig. 2e) only sulfonate is observed, in addition to $SO_{2(g)}$. The attribution of the $SO_{2(g)}$ peak was verified by applying an electrical potential to the liquid jet, such that the signal of the gas phase was smeared out and the liquid-phase signal was shifted by the amount of the applied bias voltage[26].

At pH = 0.36, with the gas-phase signal smeared out with a potential of −45 V, we identified a small peak at the high BE side of the sulfonate signal, which we assign to the solvated sulfur dioxide $SO_{2(aq.)}$ (Fig. 2f, for an alternative fit with only one species, see Supplementary Fig. 3). The $SO_{2(aq.)}$ species has binding energy very similar to gaseous $SO_2$ (174.8 eV), which is in agreement with theoretical calculations (see Table 1). The width of the peak originating from $SO_{2(aq.)}$ is 0.7 eV, which is clearly wider than the gas-phase signal (0.3 eV) but still narrower than the peaks due to the other solvated S(IV) species.

Table 1 reports the computed BEs ($BE_{theo}$) at the evGW@PBE level[27], a method which is known for reliably describing the absolute and relative XPS peak positions[28]. In soft-matter systems, such as a liquid-vapor interface, sampling various structural arrangements and solvation environments of solutes at the interface is crucial for determining reliable core-BEs and assigning meaningful statistical uncertainties to them. As outlined in the Methods section, multiple snapshots from MD trajectories were selected and optimized at the density functional theory (DFT) level before computing the BE. The $BE_{theo}$ and $\Delta BE_{theo}$ values in Table 1 corroborate experimental assignments: absolute $BE_{theo}$ positions match the experimental ones within one or two standard deviations, and the trend of BEs among different species is also well reproduced. Additionally, BE assignments in Table 1 are also supported by calculations using the GW2X method[29], detailed in Table S1 in the SI.

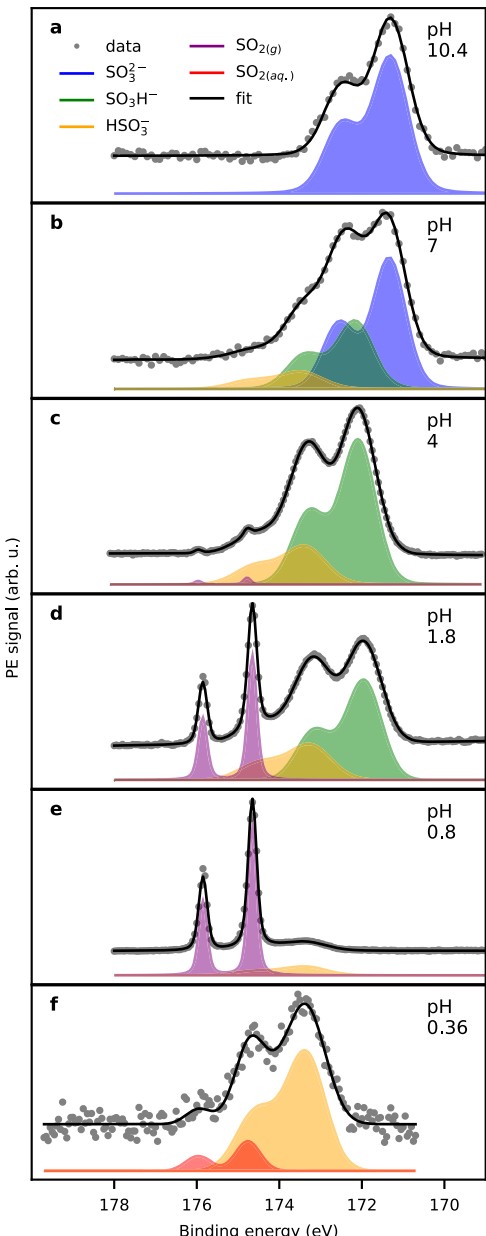

**Fig. 2 | Photoelectron spectra (S 2$p$) of aqueous solutions of 400 mM sodium sulfite.** The data are recorded with photon energies (h$\nu$) of 372 eV or 277 eV. All peaks in the S 2$p$ spectra are fitted with Gaussian doublets, except for the SO$_{2,(g)}$ peak, which was fit with a Voigt doublet. The legend is depicted in (**a**), and the fit parameters can be found in Table 1. **a** pH = 10.4, h$\nu$ = 372 eV. **b** pH = 7, h$\nu$ = 372 eV. **c** pH = 4, h$\nu$ = 372 eV. **d** pH = 1.8, h$\nu$ = 372 eV. **e** pH = 0.8, h$\nu$ = 372 eV. **f** pH = 0.36, h$\nu$ = 277 eV, −45 V bias applied to separate the gas phase.

Figure 3a summarizes the results on the equilibria of S(IV) species from MD simulations. The doubly-charged sulfite ion is the most solvated into the bulk, while the single-charged bisulfite and sulfonate species prefer to populate the subsurface, near the Gibbs dividing surface (GDS). The simulations agree with XPS measurements (see Supplementary Fig. 6), which show no significant difference in the surface propensity between the bisulfite and sulfonate species. The neutral sulfonic and sulfurous acids are surface-enhanced in the simulations. In Fig. 3a, we differentiate between SO$_2$ in the aqueous and gas phases, SO$_{2(aq.)}$ and SO$_{2(g)}$, respectively, based on the number of water oxygens around the S atom (details in the SI). We report an accumulation of SO$_2$ at the GDS, consistent with previous computational results[30]. Overall, the bulk vs. surface propensity of all the

investigated species follows a trend consistent with the Born solvation model, i.e., highly charged species are more solvated into the bulk, while neutral species display a greater surface propensity.

Figure 3b presents the free energy profile for the dissociation of sulfurous and sulfonic acid to bisulfite and sulfonate, respectively, at the liquid−vapor interface. We describe the dissociation process using the distance of the hydronium ion from the oxygen in the sulfur species, $d_h$: this metric can discriminate between undissociated acid, contact ion pairing (CIP), and solvent-separated ions, i.e., hydronium and bisulfite/sulfonate (see Discussion in the SI). Both profiles in Fig. 3b indicate a strong preference for the dissociated forms, with an almost barrierless transition from the acid ($d_h$ = 0.1) to the solvated ions ($d_h \approx$ 0.6). This likely explains why the undissociated acids have not been detected experimentally[3,17]. Moreover, an in-depth analysis of the number of water molecules coordinating the hydronium ion as a function of $d_h$ reveals how an overcoordinated state needs to be attained (near $d_h \approx$ 0.25 nm) before an ion pair can form or dissociate via solvent reorganization (see Supplementary Figs. 11−13). Interestingly, the same was previously observed in other simple electrolytes[31]. We also observe a strong CIP between sulfonate and H$_3$O$^+$: the vertical dashed black line in Fig. 3b shows that the global minimum of the free energy for sulfonic acid dissociation (solid yellow line) is, indeed, coinciding with the first solvation shell (first peak in the dashed yellow line) of the sulfur oxygens with water. This suggests that sulfonate can be stabilized at the interface by hydronium ions, while the CIP between bisulfite and hydronium ions (green lines) is weakly stable.

Panel 3c displays the free energy profiles for the interfacial dehydration of bisulfite and sulfonate. For computational reasons, the number of S−O bonds was determined from a (derivable) coordination number function, i.e., the number of contacts of the S atom with the oxygens in the system; hence sulfite and sulfonate show a minimum in $\Delta G$ at three S−O bonds, while the formation of solvated SO$_2$ occurs at a S−O bond number slightly larger than 2 (see SI for a full explanation). Bisulfite dehydration (green trace) is hindered by a barrier of approximately 20 kJ/mol, which is in agreement with the value reported for bisulfite dehydration on the surface of a well-hydrated silica surface[9]. The Fig. 3b, c show that interfacial bisulfite protonation and dehydration require very similar energies, implying that sulfurous acid may dehydrate to SO$_2$ without any further energy input at the water surface. The dehydration barrier for sulfonate (i.e., the decrease in the number of S−O bonds from 3 to 2) is approximately an order of magnitude higher than that of bisulfite. One possible explanation for this difference in barrier height is the orientation of sulfonate at the liquid−vapor interface, where the intramolecular S−H bond preferably points outward toward the gas phase (see Supplementary Fig. 14 and Discussion in the SI). This likely impedes dehydrolysis, since the intramolecular proton is not involved in hydrogen bonds with water molecules at the interface and needs to orient itself toward the aqueous bulk to facilitate dissociation.

The data in Fig. 3 indicate that sulfonate is energetically more stable at the interface compared to bisulfite, particularly at low pH, where sulfonate can be a partner in CIP. Since the tautomerization between sulfonate and bisulfite has a barrier of about 100 kJ/mol[32], sulfonate is expected to be enhanced with respect to bisulfite at low pH at the liquid-vapor interface.

We now turn our attention to the determination of the surface equilibria of the S(IV) species from S 2$p$ XPS spectra. Figure 4a shows the tautomer ratio $K_T$ (bisulfite/sulfonate) as a function of pH. The tautomer ratio around the p$K_{a2}$ (5 < pH < 8) is 3.4 ± 0.1. For comparison with the bulk value, the tautomer ratio in the bulk solution was determined from Raman spectra of sulfite solutions with a pH between 0.6 and 10 (Supplementary Fig. 7).

The average of the tautomer ratio over the range 4 < pH < 8 obtained from our Raman measurements (triangles in Fig. 4a) is 3.2 ± 0.2, i.e., in good agreement with literature values for a pH of around 4[20]. The Raman data in this pH range are also in good

agreement with our XPS measurements. With decreasing pH, both the bulk-sensitive Raman and the surface-sensitive XPS measurements show a decrease of the tautomer ratio. This effect is stronger in the surface-active XPS measurements, however.

We now utilize the tautomer ratio to evaluate the individual acid-base equilibria around the $pK_{a,2}$. Figure 4c shows the fraction of sulfite relative to the total XPS S $2p$ (cyan dots) and Raman signals (cyan triangles) as a function of pH. The solid cyan line shows the expected sulfite fraction as a function of pH based on the Henderson–Hasselbalch equation[33]:

$$x_{SO_3^{2-}} = \left( 10^{(pK_{a,2}-pH)+1} \right)^{-1}.$$ (2)

**Table 1 | Photoelectron binding energies determined from XPS experiments (S $2p$ "BE") and from evGW@PBE calculations ("BE$_{theo}$")**

| Species | Formula | BE | ΔBE | Width | BE$_{theo}$ | ΔBE$_{theo}$ |
|---|---|---|---|---|---|---|
| Sulfite | $SO_3^{2-}$ | 171.2 | 0 | 0.9 ± 0.03 | 170.9 ± 0.4 | 0 |
| Bisulfite | $SO_3H^-$ | 172.0 | 0.8 | 0.96 ± 0.02 | 172.7 ± 0.6 | 1.8 ± 0.7 |
| Sulfonate | $HSO_3^-$ | 173.4 | 2.1 | 1.15 ± 0.02 | 174.0 ± 0.3 | 3.1 ± 0.6 |
| Sulfur dioxide (aq.) | $SO_{2(aq.)}$ | 174.7 | 3.5 | 0.70 ± 0.10 | 175.9 ± 0.7 | 5.0 ± 0.8 |
| Sulfur dioxide (g) | $SO_{2(g)}$ | 174.7 | 3.5 | 0.23 ± 0.08 | 176.4 ± 1.0 | 5.5 ± 1.1 |
| Sulfate | $SO_4^{2-}$ | 173.2 | 2.0 | 0.93 ± 0.05 | 174.5 ± 2.0 | 3.6 ± 2.0 |
| Sulfurous acid | $SO_3H_2$ | | | | 177.2 ± 0.9 | 6.3 ± 1.0 |
| Sulfonic acid | $HSO_3H$ | | | | 178.4 ± 1.2 | 7.5 ± 1.2 |
| Methane sulfonate | $Me\text{-}SO_3^-$ | 173.5 | 2.3 | 0.93 ± 0.02 | | |

All values are given in eV. The measured binding energy of gaseous $SO_2$ matches the literature value[52]. In addition to sulfite solutions, we also measured sodium sulfate and methyl-sulfonate solutions as references for the sulfate and sulfonate BEs, respectively (for spectra see SI). The spin–orbit-splitting was, in all cases 1.2 eV, and the ratio of the $2p_{3/2}$ and $2p_{1/2}$ peak areas is always two. Experimental and calculated values are in agreement.

Figure 4c also shows the ratios $SO_3^{2-}/(HSO_3^- + SO_3^{2-})$ (orange) and $SO_3^{2-}/(SO_3H^- + SO_3^{2-})$ (green). From the experimentally measured tautomer ratio $K_T$ we can now determine the individual $pK_{a,2}$ of sulfite with respect to each tautomer ($pK_{a,2}^{SH}$ and $pK_{a,2}^{OH}$) by substituting either the activity $a_{SO_3H^-}$ or $a_{HSO_3^-}$ in the equation describing the $pK_a$[34]:

$$pK_{a,2} = -log\left( \frac{a_{SO_3^{2-}}}{a_{HSO_3^-} + a_{SO_3H^-}} \right).$$ (3)

This then yields Eqs. (4) and (5) for the description of the sulfite-bisulfite $pK_{a2}^{OH}$ and the sulfite-sulfonate $pK_{a2}^{SH}$ acid–base equilibria, respectively:

$$pK_{a2}^{OH} = pK_{a2} - log(1 + 1/K_T) = -log\left( \frac{a_{SO_3^{2-}}}{a_{SO_3H^-}} \right),$$ (4)

$$pK_{a2}^{SH} = pK_{a2} - log(1 + K_T) = -log\left( \frac{a_{SO_3^{2-}}}{a_{HSO_3^-}} \right).$$ (5)

Equations (3)–(5) require the knowledge of the activity coefficients $f_i$, which are, however, unknown ($a_i = f_i \cdot c_i$). We here assume $f_i = 1$, i.e., the concentration $c$ is equal to the activity $a$. Eqs. (4) and (5) then give $pK_a$ values for bisulfite and sulfonate of 6.9 and 6.3, respectively, which are used to plot the Henderson–Hasselbalch equation for the individual ions (green and orange lines in Fig. 4b, c).

The good agreement of the experimental data from XPS and Raman spectroscopy at pH values close to $pK_{a,2}$ implies that there is no measurable difference in the chemical composition of the interface and the bulk over this pH range. We now turn our attention to the pH range around $pK_{a,1}$, where surface and bulk-sensitive measurements show different results.

The fraction of the singly deprotonated S(IV) species, i.e., bisulfite and sulfonate, with respect to the concentration of dissolved $SO_{2(aq.)}$ at $pK_{a1}$ is depicted in Fig. 4b. The fraction of dissolved $SO_{2(aq.)}$ in bulk is determined directly from Raman measurements (triangles in Fig. 4b) where a distinct peak due to $SO_{2(aq.)}$ is observed. A fit of the Raman data using the Henderson–Hasselbalch equation for the equilibrium of $SO_{2(aq)}$ with the sum of sulfonate and bisulfite (cyan line) as well as for the individual cases (orange and green lines) yields a slightly lower value (1.6) for $pK_{a,1}$ than the literature value[3] (1.81), but is still in good

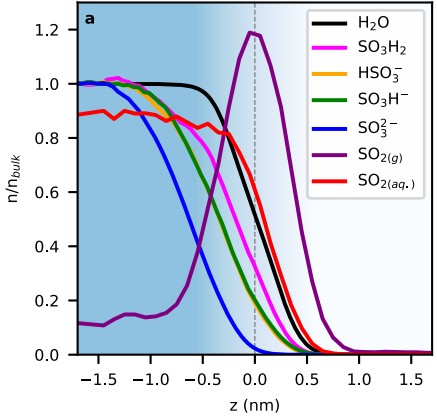
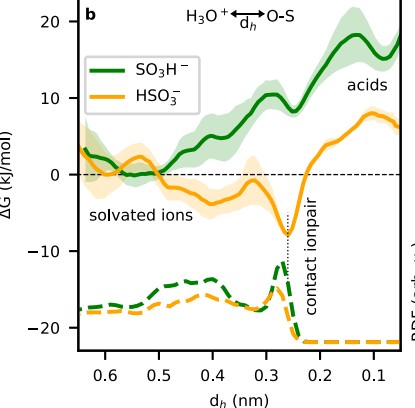
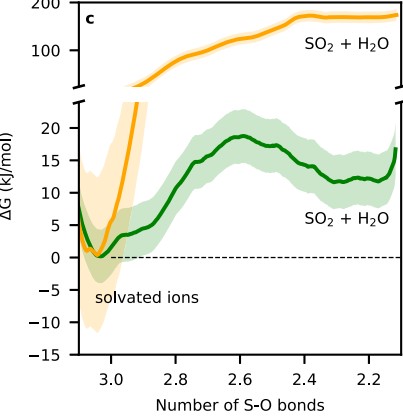

**Fig. 3 | Calculations on the acid-base equilibria of S(IV) species. a** Probability distribution profile, $n/n_{bulk}$, normalized to the bulk value as a function of the z coordinate perpendicular to the liquid–vapor interface, obtained by collecting the center-of-mass position of each species over the MD trajectory. The Gibbs dividing surface (dashed line) was calculated according to ref. 53. The blue background indicates the liquid phase and its gradual transition to the gas phase. **b** Free energy profiles for the interfacial dissociation of sulfonic (yellow) and sulfurous (green) acids as a function of the distance between the hydronium ion and the closest oxygen of the S(IV) species, $d_h$. Details on the error bar (shaded areas) can be found in section 3.5 of the Supplementary Information. Dashed traces depict the radial distribution function (RDF) between the water oxygens and the oxygen of the sulfurous species. **c** Free energy profile for the interfacial dehydration of sulfonate (yellow) and bisulfite (green) on the surface of a small water cluster, plotted against the number of S–O bonds calculated from the coordination number between the sulfur and the oxygen atom in the system. For more details including the errorbar (shaded areas) please see section 3.5 of the Supplementary Information.

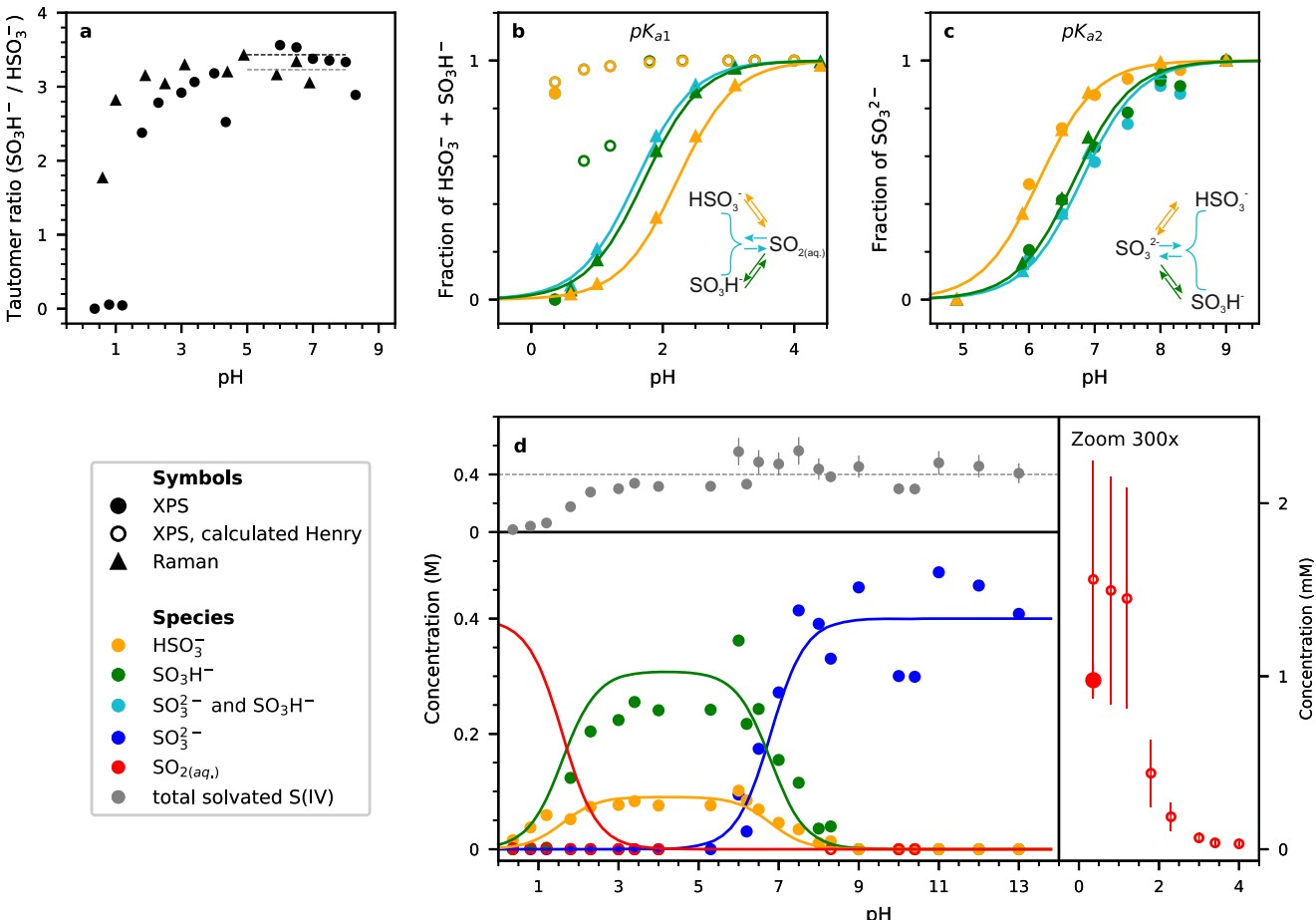

**Fig. 4 | Experimentally determined acid-base equilibria of S(IV) species.**
**a** Tautomer ratio as a function of pH. **b, c** p$K_a$s of the tautomeric forms. The horizontal lines depict the average values determined from XPS and Raman spectroscopy. **d** Concentrations of sulfur species determined via photoelectron spectroscopy. The accuracy of the pH is ±0.1. For details of the error bar of the fractions

of the ions of 0.04 and the calculation of the concentration of solvated SO$_2$, see Section 1.2.2 of the Supplementary Information. In all panels, triangles refer to Raman, full dots to photoelectron experiments, and open circles to calculated data based on photoelectron experiments.

agreement. From the Raman data at pH values around 2 (Fig. 4b), the individual bulk p$K_a$ values for the two tautomers were calculated based on a tautomer ratio of $K_T = 3.2$. For the equilibrium $SO_2 \rightleftharpoons HSO_3^-$ we determined a p$K_a$ of 2.2 and for the equilibrium $SO_2 \rightleftharpoons SO_3H^-$ a p$K_a$ of 1.7.

Figure 4b also shows the surface-sensitive XPS data (circles), which are not in agreement with the Henderson–Hasselbalch plots and the Raman measurements. This is most likely caused by the non-equilibrium conditions in the background vacuum environment (see Methods section). Nevertheless we were able to detect SO$_{2(aq.)}$ using XPS at the lowest pH in our experiments (0.36).

Figure 4d summarizes the XPS data shown in Fig. 4b, c for the whole pH range in absolute concentrations. The concentrations were calculated using two different approaches (for details, see Section 1.2 of the SI): either by comparing the peak areas of the S 2$p$ components with the O 1$s$ peak of liquid water under consideration of the respective photon fluxes and cross sections, or by comparing the O 1$s$ peak area of sulfite directly with that of liquid water, where cross-section and photon flux is equal.

The top panel in Fig. 4d displays the total concentration of all S(IV) species in the aqueous phase over the whole pH range. It is about 400 mM (the nominal bulk value) for a pH above pH 4. At pH values below 4, the concentration of solvated sulfur species decreases rapidly due to their conversion to SO$_{2(g)}$ (with some amount of SO$_{2(g)}$ already released from the time of the preparation of the solution). The right panel in Fig. 4d shows a detailed view of the pH range below 4, where the concentration of SO$_{2(aq.)}$ as determined from Henry's law (open red

circles, see Section 1.2 in the SI for details) is compared with the concentration directly measured by XPS (full red circle).

One major finding in our investigation is the strong shift of the tautomer ratio in favor of sulfonate over bisulfite at low pH, as observed in Figs. 2 and 4a. The calculations of the reaction kinetics shown in Fig. 3 indicate that this is due to the lower reaction barrier for the formation of SO$_2$ from bisulfite at the interface, in contrast to the higher barriers for the formation of SO$_2$ from sulfonate as well as for the tautomerization. The partial concentrations of the sulfur species at the interface (Fig. 4d) are of importance for the modeling of heterogeneous reactions of trace gases and other species in the atmosphere with sulfite solutions. While the partitioning of the sulfite species at the interface is close to that expected from the known bulk p$K_a$ values and tautomer ratios for pH values above 3, there is a marked enhancement of sulfonate over bisulfite at more acidic conditions. This effect is stronger in XPS than in Raman measurements, indicating that it is at least in large part a feature of the solution-vapor interface. Therefore, the changing relative abundance of the tautomers needs to be taken into account when considering the mechanisms of S(IV) oxidation. Hoffmann and Khan et al. argue that sulfonate is the main reactant for the oxidation by ozone. If this is the case more generally, the fact that sulfonate is more abundant at the surface might explain part of the surface-enhanced oxidation of S(IV) reported or suspected in previous studies[35,36].

Based on XPS and Raman spectroscopy in combination with MD simulations, we provide a quantitative assessment of the complex acid-

base equilibria of the S(IV) species at the liquid–vapor interface. We determined the tautomer ratio (bisulfite/sulfonate) over a wide pH range with bulk-sensitive Raman and surface-sensitive photoelectron spectroscopy. From these data, we determined the individual p$K_a$ values of the two tautomeric forms bisulfite and sulfonate, in the bulk solution and at the liquid–vapor interface.

Based on theoretical calculations, we provide an explanation for the enhanced depletion of bisulfite over sulfonate at the liquid–vapor interface at low pH, which is due to more favorable interfacial dehydration of bisulfite and the stabilization of sulfonate by contact ion pairing with hydronium ions. In addition, we were able to spectroscopically detect dissolved $SO_{2(aq.)}$ at the interface.

Our investigation highlights the different physio-chemical environment at interfaces vs. the bulk, which influences the mechanisms of interface vs. bulk reactions and is thus an important descriptor for the heterogeneous chemistry of aqueous aerosols, in particular in view of the multi-phase oxidation mechanisms of S(IV) species in the atmosphere, which drive, e.g., haze formation and the concentration of radicals in the atmosphere[6,8].

## Methods

### Sample preparation

Aqueous solutions of sodium sulfite (Sigma-Aldrich, >98% purity) with a concentration of 400 mM were prepared by dissolving 50.4 g in 1 L of deionized water (55 nS/cm). From the main stock solution smaller quantities (each about 200 mL) were adjusted to different pH values (measured by a pHenomenal 1100L pH meter from VWR) by drop-wise addition of diluted aqueous solutions of hydrochloric acid or sodium hydroxide under constant stirring. The pH was adjusted just prior to the measurement. In the case of solutions with low pH that produce $SO_2$ the gas phase of the sample flask was connected to a bubble flask filled with a basic solution (pH of about 10) to capture and neutralize gaseous $SO_2$.

### Photoelectron spectroscopy

The majority of the XPS measurements were performed at the P04 beamline at PETRA III (DESY, Hamburg, Germany)[37] using the EASI setup[38]. The hemispherical electron analyzer detects photoelectrons from the sample at an angle of 130° (back-scattering geometry), i.e., close to the magic angle with respect to the incident X-rays, which are circularly polarized. The diameter of the liquid microjet was 35 µm, with a flow rate of 0.7 mL/min. Since the liquid is injected into a constantly pumped vacuum chamber, the liquid phase is not in equilibrium but in a steady-state condition. The details of the experimental setup are described in ref. 38. Core-level photoelectron spectra of oxygen (O 1$s$) and sulfur (S 2$p$) were recorded at different photon energies, such that the resulting kinetic energies (and the probing depths) in both cases are identical. For example, to obtain experiments with photoelectrons having a kinetic energy of about 200 eV, we used a photon energy of 738 eV for O 1$s$ and 372 eV for S 2$p$. To explore other probing depths, we varied the photon energies accordingly to achieve kinetic energies in the range between 100 and 1000 eV. To avoid saturation of the detector, the intensity of the X-rays was attenuated in the case of O 1$s$ spectra by partially closing a baffle in the beamline.

Additional experiments were performed at the PLEIADES beamline of the SOLEIL synchrotron in Gif-sur-Yvette, France[39]. In these experiments, the flow rate of the liquid jet was 2.7 ml/min with a jet diameter of 40 µm. These measurements were performed with linearly polarized light, with the electric field vector of the incident X-rays at 55° (i.e., close to the "magic angle") with respect to the electron detection direction. Preliminary measurements were performed at the Swiss Light Source (SLS) at the surface/interface microscopy beamline (SIM) using the liquid-jet XPS endstation[40]. For the comparison of the data recorded at the different synchrotron-light facilities, we recorded the photon flux for each measurement using calibrated photodiodes.

The S 2$p$ photoelectron spectra were fitted with Gaussian (Voigt for gas phase) doublets with a 2:1 peak-area ratio ($p_{3/2}$ to $p_{1/2}$) and a spin-orbit splitting of about 1.2 eV (which is in agreement with the values for aqueous sulfate)[41], after subtraction of a combined Shirley and linear background.

### Raman spectroscopy

The sulfite solutions for the Raman spectroscopy measurements were prepared in the same way as for the XPS measurements. About 10 mL of each solution was transferred into a vial. The solution pH was measured with a Mettler Toledo pH meter Seven2Go. The Raman spectra were recorded at room temperature (24 °C) using a Kaiser Optics Raman spectrometer RXN1-785 with a 400 mW laser (785 nm) at a spectral resolution of 1 cm$^{-1}$. The spectrometer was coupled to a fiber probe, which was immersed into the solution. We assumed the same absorption cross-sections for the different sulfur species for the analysis of the Raman spectra. The peak assignment in the Raman spectra is based on previous measurements[20,21]. We fitted Raman spectra recorded in the pH range between 0.6 and 10 (for example, fits see SI) and calculated the ratios of the symmetric S−O stretch bands of sulfite, bisulfite, and sulfonate.

### Molecular dynamics and core-level calculations

The bulk vs. interfacial propensity of sulfurous acid ($SO_3H_2$), sulfonate ($HSO_3^-$), bisulfite ($SO_3H^-$) and sulfite ($SO_3^{2-}$) ions, and $SO_2$ were investigated using classical molecular dynamics (MD). Interaction parameters (i.e., the force field) were designed following the GAFF2[42] practice, which is suitable for modeling small molecules in aqueous solutions in combination with the TIP3P water[43]. The molar concentration for each of the sulfur species in the solution was 0.1 M. Sodium ions were added to neutralize the total charge of the simulation box. Polarizable effects have been introduced using a scaling-down approach of the ionic charges, which is a convenient way to include solute polarizability in nonpolarizable MD[44]. The solutes were solvated in an equilibrated liquid water slab with two open liquid–vapor interfaces, followed by 400 ns constant volume and temperature (NVT) classical MD simulation at 300 K.

First-principle molecular dynamics (FPMD) were performed to determine (a) the stability of the $SO_3H_2$ dimer, (b) sulfurous and sulfonic acid dissociation, and (c) sulfonate and bisulfite dehydration at the liquid-vapor interface. In FPMD, the forces governing the dynamics are computed "on-the-fly" using density functional theory (DFT), enabling chemical bond formation and breaking. FPMD simulations were conducted using the CP2K molecular dynamics package[45], employing PBE[46] density functional theory with dispersion correction (D3)[47]. For the free energy profiles of sulfonate and bisulfite dehydration and acid dissociation at the liquid–vapor interface, we exploited FPMD coupled with enhanced sampling methods using PLUMED[48]. Our methodology for simulating water and liquid-vapor interfaces has been extensively detailed in the literature, and comprehensive information regarding our classical and FPMD computational setups is provided in the Supporting Information (SI).

Core-level electron binding energies were calculated using the GW many-body method, which is increasingly recognized as the gold standard also for core-level predictions[28,49]. This method substantially improves the description of the correlation and exchange interactions between the electrons and the holes created upon photoemission, compared to other methods[50]. GW calculations were performed over several molecular clusters extracted from classical and FPMD trajectories, utilizing a spherical cutoff around the compound when it is at the liquid-vapor interface. We employed two different GW schemes: the eigenvalue self-consistent GW scheme (evGW)[27], starting from the PBE eigenenergies, and the GW2X[29] method from PBE0[51] eigenenergies. Table 1 reports results at the evGW level, while GW2X results are provided

in the Supplementary Information (Table S1). While evGW method provides more reliable core-level predictions in terms of absolute binding energy, the GW2X method better describes the spin−orbit coupling effect, which is relevant for the $2p_{1/2}$ and $2p_{3/2}$ line splitting. However, the latter may lead to a shift in the absolute binding energies compared to experimental values[49]. The full computational details and methodology discussion are reported in the SI.

## Data availability

Supplementary information: additional XPS spectra (valence band), details on calculations and MD simulations. The source data of the Figs. 2–4 available per Zenodo.

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

## Acknowledgements

The authors thank the beamline staff, in particular Moritz Hoesch and his team, for assistance at beamline P04 of synchrotron PETRA III. We thank the chemical safety staff at DESY for their generous support. We thank the technical service of SOLEIL for their helpful support for the work in the chemistry laboratory (project 20221549). R.S. acknowledges funding from the Swiss National Science Foundation (SNSF) project No. 200020_200306 and from the European Union's Horizon 2020 research and innovation program from the European Research Council under Grant Agreement No. 786636. M.A., A.R., and L.I. acknowledge funding from the SNSF (grant No. 188662). For high-performance computational resources and services, we acknowledge the Research Computing group at Texas A&M University in Qatar, founded by the Qatar Foundation for Education, Science and Community Development, and the use of Qatar Environment and Energy Research Institute (HBKU/QEERI) of the HPC under Project ID HPC-P21003. I.G. thanks Dr. Marcelo Carignano for insightful discussions about the simulation setup. R.D. acknowledges support from the Alexander von Humboldt Foundation through a Postdoctoral Fellowship. F.T. acknowledges funding by the Deutsche Forschungsgemeinschaft (DFG, German Research Foundation)—Project 509471550, Emmy Noether Program. F.T. and B.W. acknowledge support by the MaxWater initiative of the Max–Planck–Gesellschaft. Financial support by the Federal Ministry of Education and Research (BMBF) in the framework of the CatLab project, FKZ 03EW0015B is acknowledged.

## Author contributions

T.B., H.B., R.S., and M.A. planned the XPS experiments. T.B., S.G., C.R., R.D., C.N., F.T., and H.B. performed the XPS measurements. D.D., S.G., and T.B. performed the Raman measurements. I.G. performed the MD simulations and GW calculations and wrote the computational part of the paper. T.B. analyzed the experimental data and wrote the paper with critical feedback from all co-authors. P.A., E.P., L.I., A.R., L.A., A.B., and R.S. did preliminary XPS and Raman experiments. A.T., B.W., R.S., M.A., and H.B. supervised the project.

## Funding

## Competing interests

The authors declare no competing interests.
