## [Peer Review File · Nature Communications]

Direct observation of the complex S(IV) equilibria at the liquid-vapor interface

Corresponding Author: Dr Tillmann Buttersack

Version 0:

Reviewer comments:

Reviewer #1

(Remarks to the Author)

The authors reported very interesting and timely work on the reaction chemistry of S(IV) speciation at the liquid-vapor interface. Through a combined spectroscopic and simulation study, they have provided mechanistic insight into how different species such as bisulfite, sulfonate, sulfite, sulfurous, and sulfonic acids form/dissociate and get distributed near the interface from the bulk water phase. In particular, the bisulfite and sulfonate tautomers interconversion and their relative thermodynamic stability are clearly explained. The manuscript will have a broad appeal to both experimental and computational atmospheric chemistry and in general to speciation chemistry. I have some questions/comments (mostly minor) on the theoretical part that the authors should address to improve the manuscript before acceptance.

1. In the abstract the authors mentioned, "We also provide a molecular-scale explanation for the short lifetime of sulfurous and sulfonic acid at the liquid-vapor interface." I would suggest that the authors might want to revise this sentence or add an extra line next to it to explicitly provide the molecular explanation—why these acids are metastable at the interface?

2. The formulas for bisulfite and sulfonate in Fig. 1 are different/swapped when compared to those mentioned in the main text. For instance, on Page 3, last para, "The acids can be deprotonated twice [3], leading to the formation of the tautomers bisulfite (SO_3H^-) and sulfonate (HSO_3^-), see Fig. 1". These need to be fixed.

3. Fig. 3b caption, there are two dashed lines, but the authors only pointed to the sulfurous acid.

4. "sulforous" instead of "sulfurous" was mentioned in several places in the SI.

5. Figure 3c, I would be a little hesitant to call the y-axis as dehydration free energy. Typically, by hydration, solute-solvent hydrogen bonding is described. Here, the bisulfite or sulfonate molecules dissociate via the breaking of SO covalent bonds. This phenomenon is not exactly dehydration.

6. For the coordination number definition (in SI), the authors should mention why specific values of the cutoff distance, r_c , were chosen. Are they taken from the location of the first minimum after the first peak of radial distribution functions (RDFs)? That location is actually what defines the boundary of a coordination shell.

The choice of the powers is (10, 20)—any particular reason for choosing those? Typically, these powers determine how smooth the phase space dynamics is. The authors can make a phase space plot, $d\text{CN}/dt$ vs CN, to ensure that. Typically, a spherical (circular on a plane) or slightly-distorted spherical phase space is found, justifying the smoothness of the CN function. Also, the average value of this CN should match the CN obtained from integrating the RDF.

7. I understand that the authors wanted to save computational time, so they carved out 0.9 nm region around S from the liquid-air interface and performed FPMD on that to study acid dissociation mechanism. Do the authors truly capture the interfacial effects in this way? If they consider simulating the actual interface, how the results will be impacted? A statement on this will help the readers understand the potential shortcomings (if any) of such cluster approximation of interfacial environments.

8. The authors should provide more details on Eq. 11 in SI. Why that specific exponential was chosen? A straightforward

distance calculation can be simply hydronium oxygen and Os—why is that sophisticated expression necessary? The expression is not clearly described.

9. I think, the SI has a ton of interesting and critical info that should go to the main text to strengthen the manuscript. Particularly the 2D surfaces, S11, S12, S13—can be in the form of a composite figure and placed in the main text.

10. Figure S12; the authors can consider to replot up to 400 kJ/mol with more-dense contours to resolve the local minima they are pointing to with arrows.

11. I like Figure S13 a lot, it reveals the ion pairing/dissociation mechanism dictated by solvent reorganization. An overcoordinated state (inverted region) needs to be attained (near 0.25 nm) before an ion pair can form or dissociate via solvent reorganization. This is what is previously found as well for simple electrolytes (J. Chem. Theory Comput. 2017, 13, 8, 3470–3477).

Reviewer #2

(Remarks to the Author)

This work determined the complex S(IV) species in liquid-vapor interfaces and bulk liquid phases using XPS and Raman spectroscopy. The different S(IV) species, such as sulfite, bisulfite, sulfonate, SO₂ in gas and liquid phases is determined using XPS. Because the determination of these S(IV) species in the liquid-vapor interface is important for understanding the reaction in the aerosol, the present topic has a general interest for all the field of science. The experimental results of XPS and Raman spectroscopy are clear and are supported by theoretical calculations such as molecular dynamics simulations. However, one important concern is the importance of the XPS method for determining the tautomer ratio of S(IV) species in the present system. These ratios have been also determined by Raman spectroscopy that is the bulk-sensitive of the liquid phase and are close to those determined by XPS. The advantage of XPS measurement is surface-sensitive, and XPS determined the tautomer ratio in the liquid-vapor interface. If the behavior of the S(IV) species in the liquid-vapor interface is different from those is the bulk liquid phase, the XPS measurements is effective. But the S(IV) species is no difference between the liquid-vapor interface and bulk liquid phase. Therefore, I think that the present results are important for the field of chemistry, but do not have the general interest for all the field of science. I recommend publishing this article in more specific journal. I describe several comments below.

(1) I think that the important finding of this manuscript is the difference of fraction of SO₃⁻ and SO₃H⁻ in the pH 0.36 between XPS and Raman spectroscopy in Figure 4(b). If the authors discuss more clearly about the difference of XPS and Raman spectroscopy in this region, the present manuscript has a more general interest for the all the field of science. In the present manuscript, the authors only discussed that the discrepancy between XPS and Raman spectroscopy is caused by the non-equilibrium conditions in the background vacuum environment. Is there any possibility that the surface-sensitive XPS observes the different fraction ratio of the liquid-vapor interface compared to that of the bulk liquid phase observed by Raman spectroscopy?

(2) In Figure 2, the XPS observation of SO₂ gas in pH 1.8 and 0.8 is interesting. It is also important that SO₂ in the liquid phase is observed in pH 0.36. What is the mechanism of the dissolution of SO₂ gas in pH 0.36, which is easier than those in pH 0.8 and 1.8?

Version 1:

Reviewer comments:

Reviewer #1

(Remarks to the Author)

The authors addressed all of my comments. I am happy with the revised version. It can be accepted.

Reviewer #2

(Remarks to the Author)

The authors revised the manuscript considering the comments of the reviewers. I agree that the present manuscript has a general interest and is publishable in Nature Communications as is.

Response to the referee reports on the manuscript

”Direct observation of the complex S(IV) equilibria at the liquid-vapor interface”, under review at Nature Communications

We thank both reviewers for their positive and constructive feedback. Below we comment on the individual comments point by point. The comments by the referees are shown in black, our responses in blue, and modified text in the manuscript in red.

Reviewer 1 (Remarks to the Author): The authors reported very interesting and timely work on the reaction chemistry of S(IV) speciation at the liquid-vapor interface. Through a combined spectroscopic and simulation study, they have provided mechanistic insight into how different species such as bisulfite, sulfonate, sulfite, sulfurous, and sulfonic acids form/dissociate and get distributed near the interface from the bulk water phase. In particular, the bisulfite and sulfonate tautomers interconversion and their relative thermodynamic stability are clearly explained. The manuscript will have a broad appeal to both experimental and computational atmospheric chemistry and in general to speciation chemistry. I have some questions/comments (mostly minor) on the theoretical part that the authors should address to improve the manuscript before acceptance.

1. In the abstract the authors mentioned, “We also provide a molecular-scale explanation for the short lifetime of sulfurous and sulfonic acid at the liquid-vapor interface.” I would suggest that the authors might want to revise this sentence or add an extra line next to it to explicitly provide the molecular explanation—why these acids are metastable at the interface?

We thank the referee for the suggestion and have modified the above sentence to:

”On a molecular-scale level, we explain this with the formation of a stable contact ion pair between sulfonate and hydronium ions, and with the higher energetic barrier for the dehydration of sulfonic acid at the liquid-vapor interface.”

2. The formulas for bisulfite and sulfonate in Fig. 1 are different/swapped when compared to those mentioned in the main text. For instance, on Page 3, last para, “The acids can be deprotonated twice [3], leading to the formation of the tautomers bisulfite (SO_3H^-) and sulfonate (HSO_3^-), see Fig. 1”. These need to be fixed.

We thank the referee for spotting this. We corrected this typo in Figure 1, and in the process also a second typo which we found during revision (swapping labeling of arrows). In addition we modified the cited sentence to:

”The acids can be deprotonated twice [3], **in a first step leading to the formation of the tautomers bisulfite (SO_3H^-) and sulfonate (HSO_3^-), and in a second step to form sulfite (SO_3^{2-})**, see Fig. 1.”

Fig. 1 Updated version of Figure 1

3. Fig. 3b caption, there are two dashed lines, but the authors only pointed to the sulfurous acid.

We have updated Figure 3b and we modified the text in the main manuscript as follow:

We also observe a strong CIP between sulfonate and H_3O^+ : the vertical dashed black line in Figure 3b shows that the global minimum of the free energy for sulfonic acid dissociation (solid yellow line) is, indeed, coinciding with the first solvation shell (first peak in the dashed yellow line) of the sulfur oxygens with water. This suggests that sulfonate can be stabilized at the interface by hydronium ions, while the CIP between bisulfite and hydronium ion (green lines) is weakly stable.

Fig. 2 Updated version of Figure 3

4. “sulforous” instead of “sulfurous” was mentioned in several places in the SI.

We have corrected these typos in the revised version of the manuscript.

5. Figure 3c, I would be a little hesitant to call the y-axis as dehydration free energy. Typically, by hydration, solute-solvent hydrogen bonding is described. Here, the bisulfite or sulfonate molecules dissociate via the breaking of SO covalent bonds. This phenomenon is not exactly dehydration.

We thank the referee for the comment but respectfully disagree that the term “dehydration” was used in the wrong context in the manuscript. The term *dehydration* is indeed used to describe the removal of water molecules from the solvation shell of solutes in aqueous solution. However, more generally the term *dehydration* is also applied for reactions of the type $A \rightarrow B + H_2O$. For references see, e.g., Manabe et al., *Dehydration Reactions in Water. Brønsted Acid-Surfactant-Combined Catalyst for Ester, Ether, Thioether, and Dithioacetal Formation in Water*, J. Am. Chem. Soc. 124, 11971-11978 (2002), and Wang et al., *Comprehensive Study of the Hydration and Dehydration Reactions of Carbon Dioxide in Aqueous Solution*, J. Phys. Chem. A 114, 1734-1740, (2010).

6. For the coordination number definition (in SI), the authors should mention why specific values of the cutoff distance, r_c , were chosen. Are they taken from the location of the first minimum after the first peak of radial distribution functions (RDFs)? That location is actually what defines the boundary of a coordination shell. The choice of the powers is (10, 20)—any particular reason for choosing those? Typically, these powers determine how smooth the phase space dynamics is. The authors can make a phase space plot, dCN/dt vs CN, to ensure that. Typically, a spherical (circular on a plane) or slightly-distorted spherical phase space is found, justifying the smoothness of the CN function. Also, the average value of this CN should match the CN obtained from integrating the RDF.

The r_c values were not taken as the location of the first minimum of the RDF in Figure 3b. Here, we followed the approach of Pietrucci and Saitta (PNAS 112, 49, 2015, <https://www.pnas.org/doi/full/10.1073/pnas.1512486112>) and of Sinopoli et al. (JPC-B, 125, 2021, <https://pubs.acs.org/doi/10.1021/acs.jpcc.1c0166>) where the coordination number, CN, is used to identify atoms that are chemically bonded: r_c and the (p,q) exponents in Table S2 were adjusted to better describe the bonding formation and breaking.

For example, in equation 10 of SI the coordination number $n(r_{i,j})$ identifies the number of hydrogen atoms chemically bonded to the oxygens of the acid (i.e., sulfurous or sulfonic acids): with our choice of r_c and of the (p, q) exponent, n is 0.9 whenever an H atom is chemically bonded to the oxygen atom of the acids, O_s , and ranges between 0.1 and 0.2 when the H is within the first solvation shell of the oxygens. Similarly, the r_c and the (p, q) exponents used in the dehydration study (i.e., section 3.6 of SI) have been tuned to get a CN of approximately 0.9 for each S-O bond.

The tuning of these parameters has been performed by using the NVT unbiased runs performed prior to running the MTD runs, and by visual inspection of both bias and unbiased MD simulations.

Here, we report the n_a vs. $d(n_a)/dt$ collected over the MTD run for the sulfurous acid dissociation (i.e., from Fig. S11). As the reviewer pointed out in his/her comment a smooth CN should result in a spherical (circular on a plane) or slightly spherical phase space: in our case, the plot looks like a spiral because the collective variable, n_a , is biased during the run.

Fig. 3 The coordination number n_a , (i.e., equation 10 from SI) vs. $d(n_a)/dt$ obtained from the OPES-MTD trajectory for the sulfurous acid dissociation.

We expanded the description in the SI (section 3.5) as follow:

Here, the coordination number defined in Eq. 9 and Eq. 10 has been used to identify H atoms chemically bonded to the oxygen atoms, O_s , of the acids. For each H atom chemically bonded with one O, n assumes a value of approximately 0.9. n is approximately 0.2 when a H atom is within the first solvation shell of one O, while

n values smaller than 0.2 correspond to H atom beyond the first solvation shell (i.e, H_3O^+ solvent-separated from the conjugated ion). In this way, $n_a \approx 1.8$ for sulfurous acid and ≈ 0.9 for bisulfite. Similarly, n_a is approximately 0.9 for sulfonic acid and ranges from 0.1 to 0.2 for sulfonate ions.

7. I understand that the authors wanted to save computational time, so they carved out 0.9 nm region around S from the liquid-air interface and performed FPMD on that to study acid dissociation mechanism. Do the authors truly capture the interfacial effects in this way? If they consider simulating the actual interface, how the results will be impacted? A statement on this will help the readers understand the potential shortcomings (if any) of such cluster approximation of interfacial environments.

We thank the reviewer for this comment and we agree that this point deserves further explanations. The study of chemical reaction by FPMD and enhanced sampling methods require to carefully choose the size, especially when the reaction barriers and chemical mechanisms are unknown. Moreover, the computational cost of FPMD limits the possibility of tuning our simulation setup by trial and error. In this work we explore 4 unknown chemical reactions (i.e., the deprotonation of sulfurous and sulfonic acid, and their dehydration): for this reason and contrary to our previous computational work employing liquid water slabs, we here preferred to adopt a cluster approach.

In selecting the cluster size, we started from our and other previous work on the vapor-liquid water interface. For example, Baer et al., JCP 135, 124712 (2011) revised the properties of the vapor-liquid water interface by FPMD: based on the density profile the thickness of the interface environment (identification by the starting of a flat bulk density profile, Figure 1 in Baer et al.) is approximately 0.5 nm at 300 K, with slight variation depending on the DFT method used. In our

acid dissociation study, we carved out a 0.9 nm snapshot centered to the S atom from the classical MD trajectory: looking at the RDF in Figure 3b it is possible to see a saturation of the RDF at 0.6 nm from the interfacial S atom, as a sign of a bulk like behavior. As explained in the SI, this cluster size enables the study the deprotonation process including solvent separated ions. As a final remark, the snapshots were extracted from the 400 ns trajectory classical MD of an air-liquid water interface (section 3.1 of SI). In the timeframe of our FPMD (≈ 10 ps), the cluster does not have enough time to rearrange in a configuration that significantly differs from that at the interface.

We added the following in section 3.5 of the SI: Due to the cost of simulating multi reaction mechanisms at the air-liquid water interface by FPMD, here we adopted a cluster approximation of the interfacial environment, i.e., we used snapshot carved from the classical MD trajectory of the solute at the air-liquid water interface as starting configurations of our FPMD. The rationale of this choice is that in the timeframe of our FPMD (≈ 10 ps), clusters of sufficient size do not have enough time to rearrange in a configuration that significantly differ from that at the interface.

The starting configurations were extracted from the classical MD trajectory with sulfurous (or sulfonic) acid solvated at the air-liquid water interface using a spherical cutoff radius of 0.9 nm centered on the S atom. The cutoff radius was larger than the radius of the second solvation shell between the oxygen atoms of the acids and the water oxygens (see Figure 3b of main manuscript). Moreover, Baer et al.,REF[JCP 135, 124712 (2011)] has revised the properties of the vapor-liquid water interface by FPMD reporting a thickness of the interface environment of approximately 0.5 nm at 300 K, smaller than the cutoff here adopted (i.e., 0.9 nm). Indeed, looking at the radial distribution function (RDF) between the oxygen of the sulfurous species and the water oxygen in Figure 3b, we observe a saturation of the RDF at ≈ 0.6 nm from the interfacial S atom, as a sign of a bulk

like behavior. Thus, we are confident that our carved clusters are reliable initial configurations comprising of a single sulfurous (or sulfonic) acid solvated in an interfacial environment with at least two solvation shells.

Furthermore, we added in section 3.6 of the SI:

The starting configuration for the MTD runs was extracted from the classical MD trajectory with sulfonate (or bisulfite) using a spherical cutoff radius of 0.6 nm centered on the S atom, resulting in a water cluster of 12 water molecules with sulfonate (or bisulfite) solvated on top of it. While a cluster approach is an unavoidable approximation of the interfacial environment due to the computational cost of FPMD, a cluster size of 0.6 nm is large enough to catch the solute in an environment resembling those of the air-liquid water interface. Also keeping in mind that in the timeframe of our FPMD (≈ 10 ps), clusters of sufficient size do not have enough time to rearrange in a configuration that significantly differ from that at the interface from where they were carved. (see also mentioned in section 3.5)

8. The authors should provide more details on Eq. 11 in SI. Why that specific exponential was chosen? A straightforward distance calculation can be simply hydronium oxygen and Os—why is that sophisticated expression necessary? The expression is not clearly described.

The sophisticated expression comes from the needs of (a) capturing the position of the hydronium ion since the excess proton diffuses by Grotthuss in water, and (b) to have a collective variable that is a derivable function of the atomic coordinates for the MTD. Our CV was inspired by the previous work of Park et al. on acetic acid dissociation [55], and here we biased our FPMD using PLUMED plug-in: Eq. (11) describes the way we have found to implement the distance between the hydronium oxygen and the closest Os in PLUMED. There might be other ways to implement

this distance, e.g., keeping the constraints (a) and (b) mentioned above. We would appreciate recommendations by the reviewer to consider those for future work.

We expanded the previous text in SI as follow:

Since in water the excess proton can diffuse by Grotthuss mechanism in water, we identify the hydronium oxygen as the water oxygen, O_w , coordinated with three hydrogens. Then, the distance between the hydronium oxygen and the closest O_s of the acid was defined as

$$d_h = f(n_a) \frac{\sum_{i \in O_w} r_i e^{\lambda n_i}}{\sum_{i \in O_w} e^{\lambda n_i}} \quad (1)$$

where r_i is the distance of O_w from the closest O_s of the acid, while n_i is the number of H atoms coordinated with O_w .

The distance defined above is a derivable function of the atomic coordinates, which makes it suitable to be used in combination with OPES-MTD. For sufficiently large λ , the fraction term on the right side of Eq. 11 converges to the distance, d_h , between the three-hydrogen coordinated O_w (i.e., the hydronium ion) and the closest O_s . The pre-factor $f(n_a) = (1 - (n_a/n_c)^p)/(1 - (n_a/n_c)^q)$ is a switching function that sets d_h to zero when the acid is protonated.

d_h and n_a are two derivable functions of the atomic coordinates. While n_a describes the protonation/deprotonation of the acid, d_h is necessary to distinguish between contact ion pairing (CIP) configurations (i.e., when the hydronium ion is still in contact with the deprotonated base) and dissociated products (i.e., when the base and the hydronium ion are solvent-separated). Table S2 contains the parameters utilized for d_h and n_a . These CVs and parameters were inspired by previous literature reports on acetic acid dissociation[55] and tuned for our 5ps NVT unbiased MD runs.

9. I think, the SI has a ton of interesting and critical info that should go to the main text to strengthen the manuscript. Particularly the 2D surfaces, S11, S12, S13—can be in the form of a composite figure and placed in the main text.

We appreciate the positive feedback of the referee in regard to Figures S11-S13. We have updated the figures to correct the typos (see response to query No. 2). We think this comment is linked to query No. 11 of the same reviewer, where the reviewer pointed about the work of Roy et al. (J. Chem. Theory Comput. 2017, 13, 8, 3470–3477). While our 2D maps Fig. S11-S12 capture some of the insights pointed out by Roy et al., the 2D maps of Roy et al. were obtained by classical MD and thus, they were much more sampled and they can more safely interpreted as free energy maps. In our case, as mentioned in the SI, *“due to inherent sampling limitations within current FPMD, Figure S11 and S12 should be interpreted as a 2D bias landscape, i.e., regions that were most frequently visited and, thus, bias potential deposited, rather than a fully sampled 2D free energy map. Nonetheless, these 2D bias landscapes are highly valuable for extracting information about potential new pathways and intermediates.”*

For this reason and to maintain the overall length and succinctness of the manuscript we would like to keep the original number and layout of the figures in the main manuscript. Nevertheless, we added a comment to draw more attention to Figs. S11-S13 in the main manuscript and to the work of Roy et a. (see query No. 11)

10. Figure S12; the authors can consider to replot up to 400 kJ/mol with more-dense contours to resolve the local minima they are pointing to with arrows.

We thank the reviewer for this suggestion and replotted Fig. S12 accordingly.

Fig. 4 Updated version of Figure S12.

11. I like Figure S13 a lot, it reveals the ion pairing/dissociation mechanism dictated by solvent reorganization. An overcoordinated state (inverted region) needs to be attained (near 0.25 nm) before an ion pair can form or dissociate via solvent reorganization. This is what is previously found as well for simple electrolytes (J. Chem. Theory Comput. 2017, 13, 8, 3470–3477).

We appreciate the positive feedback and the suggestion to include this reference.

We added the following in the main manuscript (end of page 6):

Moreover, an in-depth analysis of the number of water molecules coordinating the hydronium ion as a function of d_h reveals how an overcoordinated state needs to be attained (near $d_h \approx 0.25$ nm) before an ion pair can form or dissociate via solvent reorganization (see Figs. S11 to S13 in the SI). Interestingly, the same was previously observed in other simple electrolytes (J. Chem. Theory Comput. 2017, 13, 8, 3470–3477).

Reviewer 2 (Remarks to the Author):

This work determined the complex S(IV) species in liquid-vapor interfaces and bulk liquid phases using XPS and Raman spectroscopy. The different S(IV) species, such as sulfite, bisulfite, sulfonate, SO₂ in gas and liquid phases is determined using XPS. Because the determination of these S(IV) species in the liquid-vapor interface is important for understanding the reaction in the aerosol, the present topic has a general interest for all the field of science. The experimental results of XPS and Raman spectroscopy are clear and are supported by theoretical calculations such as molecular dynamics simulations. However, one important concern is the importance of the XPS method for determining the tautomer ratio of S(IV) species in the present system. These ratios have been also determined by Raman spectroscopy that is the bulk-sensitive of the liquid phase and are close to those determined by XPS. The advantage of XPS measurement is surface-sensitive, and XPS determined the tautomer ratio in the liquid-vapor interface. If the behavior of the S(IV) species in the liquid-vapor interface is different from those is the bulk liquid phase, the XPS measurements is effective. But the S(IV) species is no difference between the liquid-vapor interface and bulk liquid phase. Therefore, I think that the present results are important for the field of chemistry, but do not have the general interest for all the field of science. I recommend publishing this article in more specific journal. I describe several comments below.

1. I think that the important finding of this manuscript is the difference of fraction of SO₃⁻ and SO₃H⁻ in the pH 0.36 between XPS and Raman spectroscopy in Figure 4(b). If the authors discuss more clearly about the difference of XPS and Raman spectroscopy in this region, the present manuscript has a more general interest for the all the field of science. In the present manuscript, the authors only discussed that the discrepancy between XPS and Raman spectroscopy is caused by the non-equilibrium conditions in the background vacuum environment. Is there

any possibility that the surface-sensitive XPS observes the different fraction ratio of the liquid-vapor interface compared to that of the bulk liquid phase observed by Raman spectroscopy?

We thank the reviewer for this comment. As pointed out by the referee and also at several places in the manuscript, Raman spectroscopy is a bulk sensitive method, while XPS is a surface sensitive method, with a probing depth of a few nanometers under the experimental conditions used in the present measurements (see, e.g., the second sentence in the results section on page 4 of the main manuscript: "*The estimated probing depth into the solution in the XPS experiments is of the order of 2 nm.*" or in the conclusion: "*(...) with bulk-sensitive Raman and surface-sensitive photoelectron spectroscopy.*")

We have observed and discussed that - in contrast to the bulk - at the liquid-vapor interface only sulfonate (and no bisulfite) is observed at the lowest pH values in our measurements (see e.g., page 5: "*In addition, the relative intensity of sulfonate vs. bisulfite changes in favor of sulfonate, until in the spectrum at $\text{pH} = 0.8$ (Fig. 2e) only sulfonate is observed, in addition to $\text{SO}_{2(g)}$.*" and at the end of page 10 "*With decreasing pH both the bulk-sensitive Raman and the surface-sensitive XPS measurements show a decrease of the tautomer ratio. This effect is stronger in the surface-active XPS measurements, however.*")

In addition to the experimental part, the manuscript contains state-of-the-art calculations to explain the finding (see Figure 3): 1) The sulfonate ion forms a stable contact ion pair at the interface, and 2) The free energy barrier for dehydration of sulfonic acid (metastable protonated form of sulfonate) is larger than for sulfurous acid (see bottom of page 8: "*This suggests that sulfonate can be stabilized at the interface by hydronium ions. (...) , implying that sulfurous acid may dehydrate to SO_2 without any further energy input at the water surface. The dehydration barrier for sulfonate (i.e., the decrease in the number of S-O bonds from 3 to 2)*")

is approximately an order of magnitude higher than that of bisulfite. One possible explanation for this difference in barrier height is the orientation of sulfonate at the liquid-vapor interface, where the intramolecular S-H bond preferably points outward towards the gas phase (see Fig. S14 and discussion in the SI). (...))

Also in the conclusion, we have already highlighted a possible explanation for depletion of bisulfite from the liquid-vapor interface: *"Based on theoretical calculations we provide an explanation for the enhanced depletion of bisulfite over sulfonate at the liquid-vapor interface at low pH, which is due to a more favorable interfacial dehydration of bisulfite and the stabilization of sulfonate by contact ion pairing with hydronium ions."*

2. In Figure 2, the XPS observation of SO₂ gas in pH 1.8 and 0.8 is interesting. It is also important that SO₂ in the liquid phase is observed in pH 0.36. What is the mechanism of the dissolution of SO₂ gas in pH 0.36, which is easier than those in pH 0.8 and 1.8?

We performed the measurement of dissolved SO_{2(aq.)} at a very low pH (0.36) to maximize its relative abundance. At a given SO₂ partial pressure in the gas phase, the concentration of SO_{2(aq.)} is the same, independent of pH. However, at higher pH, the different bisulfite/sulphurous species are dominating. Thus, performing this measurement at very low pH ensures that SO_{2(aq.)} is the only/dominant species and can be identified spectroscopically.